# Bedside Selection of Positive End Expiratory Pressure by Electrical Impedance Tomography in Patients Undergoing Veno-Venous Extracorporeal Membrane Oxygenation Support: A Comparison between COVID-19 ARDS and ARDS from Other Etiologies

**DOI:** 10.3390/jcm11061639

**Published:** 2022-03-16

**Authors:** Michela Di Pierro, Marco Giani, Alfio Bronco, Francesca Maria Lembo, Roberto Rona, Giacomo Bellani, Giuseppe Foti

**Affiliations:** 1School of Medicine and Surgery, University of Milano-Bicocca, 20900 Monza, Italy; m.dipierro3@campus.unimib.it (M.D.P.); marco.giani@unimib.it (M.G.); f.lembo2@campus.unimib.it (F.M.L.); giacomo.bellani1@unimib.it (G.B.); giuseppe.foti@unimib.it (G.F.); 2Department of Emergency and Intensive Care, Azienda Socio Sanitaria Territoriale Monza, 20900 Monza, Italy; r.rona@asst-monza.it

**Keywords:** COVID-19, ECMO, mechanical ventilation, ARDS, acute respiratory distress syndrome

## Abstract

Background: The interest in protective ventilation strategies and individualized approaches for patients with severe illness on veno venous extracorporeal support has increased in recent years. Wide heterogeneity exists among patients with COVID-19 related acute respiratory distress syndrome (C-ARDS) and ARDS from other etiologies (NC-ARDS). EIT is a useful tool for the accurate analysis of regional lung volume distribution and allows for a tailored ventilatory setting. The aim of this work is to retrospectively describe the results of EIT assessments performed in patients C-ARDS and NC-ARDS undergoing V-V ECMO support. Methods: A clinical EIT-guided decremental PEEP trail was conducted for all patients included in the study and mechanically ventilated. Results: 12 patients with C-ARDS and 12 patients with NC-ARDS were included in the study for a total of 13 and 18 EIT evaluations, respectively. No significant differences in arterial blood gas, respiratory parameters, and regional ventilation before and after the EIT exam were recorded. The subset of patients with NC-ARDS whose EIT exam led to PEEP modification was characterized by a lower baseline compliance compared with the C-ARDS group: 18 (16–28) vs. 27 (24–30) (*p* = 0.04). Overdistension significantly increased at higher steps only for the NC-ARDS group. A higher percentage of overdistension was described in patients with NC-ARDS when compared with patients with C-ARDS. Conclusions: EIT is feasible in patients with COVID-19-associated ARDS on veno-venous extracorporeal support and may help in tailoring the PEEP setting. Overall, severe COVID-19-related ARDS presents respiratory characteristics similar to severe “classical” NC-ARDS. However, C-ARDS is associated with a lower risk of overdistension at a higher PEEP level compared with NC-ARDS.

## 1. Introduction

During the recent SARS-CoV-2 pandemic, Veno-venous Extracorporeal Membrane Oxygenation (V-V ECMO) has been used as a rescue treatment for severe COVID-19 related acute respiratory distress syndrome (ARDS) [1,2,3,4]. Its main indications are severe hypoxemia or hypercapnia in the case of a failure in conventional therapy, which is based on a lung protective mechanical ventilation strategy, with optimized positive end expiratory pressure (PEEP) [5].

In recent years, the interest in the development of individualized ventilation strategies in patients who are critically ill has increased [6].

As part of a lung protective individualized ventilation strategy, electrical impedance tomography (EIT) proved to be a useful tool for guiding mechanical ventilation and for PEEP titration both in COVID-related ARDS (C-ARDS) and ARDS from other etiologies (NC-ARDS), allowing us to minimize both lung overdistension and collapse [7,8]. The reduction in regional overdistension and/or the recruitment of the collapsed region may allow us to reduce local stress and strain, thus limiting ventilation-induced lung injury (VILI). In particular, patients with C-ARDS could have a specific pathophysiological phenotype compared with those affected by NC-ARDS 19. An intermediate PEEP level seemed appropriate in almost half of patients with C-ARDS [9], but wide heterogeneity exists [10]. Similarly, a broad variability in optimal PEEP has been observed also in patients with severe NC-ARDS receiving ECMO support (a specific situation in which tidal volume and minute ventilation are markedly reduced) [11,12], thus confirming the rationale for individualized PEEP setting and reinforcing the need for personalized titration of ventilation settings [6,7,8,9,10,11].

In our intensive care unit (ICU), EIT has been widely used for research purposes [13,14,15,16,17,18]. A recent paper described our clinical experience using EIT to individualize and optimize ventilation management in patients with critical NC-ARDS (with or without ECMO) [19]. More recently, during the outbreak of SARS-CoV-2-related respiratory failure, the use of EIT to set PEEP was routinely implemented [9,10]. The aim of this work is to retrospectively compare the results of EIT assessments performed in patients with C-ARDS and NC-ARDS undergoing V-V ECMO support. Specifically, we compared respiratory mechanics, regional ventilation distribution, and optimal PEEP levels in these populations.

## 2. Materials and Methods

This is a comparison study conducted in the General Intensive Care Unit (ICU) at “San Gerardo Hospital” in Monza (Italy), a tertiary level centre for the treatment of ARDS and ECMO. Data of patients with C-ARDS were collected as part of the STORM study (Spallanzani Institute approval number 84/2020; NCT04424992), which did not include the present evaluation as a pre-specified endpoint. Data obtained in patients with NC-ARDS for a recent retrospective study were used for comparison [19]. Informed consent was waived.

### 2.1. Data Collection

Data of 12 patients with NC-ARDS on ECMO support were retrieved (as a subset of the original 41 patients) from the database of our previous study [19], which included all patients who underwent a clinical EIT-guided PEEP trial in our general ICU from January 2017 to December 2019. For the C-ARDS population, we included all adult patients with ARDS who were mechanically ventilated under ECMO support with a confirmed diagnosis of COVID-19 admitted to our 10-bed ICU between January 2021 and May 2021 and who underwent an EIT examination to set PEEP. EIT analysis was performed in both situations (patients with C-ARDS and NC-ARDS) upon clinical indication by the treating physicians. No exclusion criteria were applied. Patient characteristics at baseline and clinical features before and after the EIT assessment were retrospectively obtained in both groups from electronic medical records. To adequately describe patient oxygenation while excluding the confounding effect of extracorporeal gas exchange, the pulmonary shunt fraction [20] was calculated. Mixed venous blood was sampled from a pulmonary artery catheter, which is routinely used at our institution to monitor patients on ECMO.

### 2.2. EIT Examination Protocol

The EIT examination protocol was the same for both groups. All patients were in supine position, deeply sedated and paralyzed. An EIT belt equipped with 16 electrodes was placed around the patient’s chest at the fourth or fifth intercostal space at the parasternal line and connected to an EIT monitor (Pulmovista^®^ 500, Dräger Medical GmbH, Lübeck, Germany). EIT images were continuously recorded at 20 Hz. The resulting images and waveforms identified four ventral-to-dorsal horizontal regions of interest (ROI).

According to our clinical practice, we approached PEEP setting with a decremental PEEP trial, starting from a PEEP level 2 to 4 cm H_2_O above the pre-existing PEEP level. As previously described [19], the decremental PEEP trial was preceded by a recruitment manoeuvre (35 cm H_2_O for 30 s), followed by 2 cm H_2_O decremental steps lasting 10 min each. At each PEEP level, mechanical respiratory parameters and regional EIT data (i.e., tidal volume distribution in the ventral to dorsal regions of interest) were recorded. Specifically, we calculated the driving pressure as the difference between plateau pressure and total PEEP, and the respiratory system compliance, computed as the ratio between expired tidal volume and driving pressure. These were followed by a more detailed analysis to evaluate the functional EIT images with dedicated software (Diagnostic View, Pulmovista^®^ 500, Drager Medical GmbH, Lubeck, Germany). The percentage of overdistension (compliance loss at higher PEEP, CL HP) and the percentage of collapse (compliance loss at lower PEEP, CL LP) were displayed at each PEEP level as described by Costa et al. [21]. The inspiratory fraction of oxygen (FiO_2_) was left unchanged throughout the PEEP trial. A recruitment manoeuvre (35 cm H_2_O for 30 s) followed the lowest step of the decremental PEEP trial. We defined the PEEP value set by clinicians before the EIT study as PEEP_PRE_, whereas the new PEEP level was selected by clinicians as PEEP_POST_. PEEP_POST_ was set at the PEEP level, which allowed the best compromise between alveolar overdistension and collapse at the EIT evaluation. When the EIT findings did not allow us to unequivocally decide between two PEEP levels because of slight differences between two PEEP levels, the PEEP_POST_ setting was left to the decision of the clinician, who integrated the EIT exam results and other patient information such as hemodynamics and oxygenation status. 

### 2.3. Statistical Analysis

Categorical variables were expressed as absolute (relative) frequency. Continuous variables were synthesized as median [interquartile range, IQR], as appropriate. The Wilcoxon test was used to compare the clinical characteristics of patients on ECMO with C-ARDS and NC-ARDS. The Wilcoxon signed rank test was used to compare changes in respiratory and EIT parameters before and after the PEEP change and to compare overdistension and de-recruitment at different PEEP levels. Overdistension and de-recruitment were computed using PEEPpost as a reference. The Bonferroni correction was used to adjust for multiple comparisons. A *p*-value < 0.05 (two-tailed) was considered statistically significant. Statistical analysis was performed with the JMP 15 software (SAS, Cary, NC, USA). A *p*-value < 0.05 (two-tailed) was considered statistically significant. Statistical analysis was performed with the JMP 15 software (SAS, Cary, NC, USA).

## 3. Results

Between January 2021 and May 2021, an EIT decremental PEEP trial was performed, following clinical indication, 23 times in 22 patients with a diagnosis of COVID-19 related ARDS (C-ARDS); 12 of them (55%) were on veno-venous ECMO support at time of evaluation and were included in the study. The exam was performed twice in one patient for a total of 13 studies. The control group included patients admitted between January 2017 and December 2019 with ARDS from causes other than SARS-CoV-2 infection (NC-ARDS): out of the original subset of patients, a total of 26 decremental PEEP trials were performed in 18 patients. We limited the current analysis to the 12 patients who were on veno-venous ECMO support at the time of EIT exam for a total 18 EIT evaluations.

The demographic and clinical characteristics are summarized in Table 1. Patients with C-ARDS had higher Body Mass Index (BMI) compared with patients with NC-ARDS. The duration of mechanical ventilation, ECMO support, and ICU stay did not differ between the two groups, whereas ICU and hospital mortality were higher in the NC-ARDS group (*p* < 0.05).

Median time from intubation to EIT exam was similar in both groups (4 days [IQR 2–9] for C-ARDS group vs. 5 [IQR 3–12] for the control group, *p* = 0.35). Clinical characteristics and ventilation parameters before the EIT examination are detailed in Table 2. All patients were ventilated in volume-controlled mode with ultra-protective ventilation strategy. ARDS severity, as indicated by pulmonary shunt fraction, compliance, and respiratory parameters, was similar in the two study populations.

### 3.1. PEEP Trial

In the C-ARDS population, the tested PEEP levels ranged from 9 to 24 cm H_2_O, and in the NC-ARDS group, the tested PEEP levels ranged from 6 to 21 cm H_2_O in 2 cm H_2_O steps.

In patients with C-ARDS, PEEP was left unchanged after the EIT exam in six cases (46%). Conversely, PEEP was increased in five exams (38%), with a maximum variation of 2 cm H_2_O, and reduced in two exams (15%), with a maximal variation of 2 cm H_2_O.

Similarly, in the NC-ARDS group, PEEP was left unchanged in five evaluations (28%); increased in five evaluations (28%), with maximal variation of 4 cm H_2_O; and decreased in eight evaluations (44%), with maximal variation of 3 cm H_2_O.

Table 3 shows the respiratory settings, blood gas, and respiratory mechanics parameters and the distribution of ventilation (EIT) before and after the EIT exam. No significant differences were recorded in the arterial blood gas parameters before and after the EIT exam in any of the study groups. Minor differences in the regional distribution of tidal volume were recorded.

When considering only the subset of exams that led to PEEP modification, the baseline compliance of the NC-ARDS group was significantly lower compared with the C-ARDS group, 18 [IQR 16–28] vs. 27 [IQR 24–30] (*p* = 0.04), but became similar to that in patients with C-ARDS after PEEP adjustment, 26 [IQR 19–39] vs. 28 [IQR 22–33] (*p* = 0.5).

### 3.2. Overdistension and Derecruitment 

At the end of the EIT exam, the percentages of overdistension (CL HP) and de-recruitment (CL LP) at each PEEP level were recorded. Figure 1 shows the trends of overdistension and de-recruitment at each step for each study populations, considering PEEP_POST_ as the reference. De-recruitment increased significantly at lower PEEP levels and reduced significantly at higher PEEP levels in both populations (*p* < 0.05). Overdistension significantly increased at higher steps (both steps +2 and +4) for the NC-ARDS group, albeit the increase was not statistically significant in the C-ARDS group. Overdistension significantly decreased only at the lowest step compared with PEEP_POST_ in both populations.

The trends in overdistension and de-recruitment were similar in both groups. However, a higher percentage of overdistension was described in patients with NC-ARDS when compared with patients with C-ARDS (*p* < 0.05).

## 4. Discussion

In the present study, we retrospectively evaluated the clinical application of EIT to set positive end-expiratory pressure in patients with COVID-19-related ARDS and we compared our findings with those obtained from a previously published study in patients with ARDS due to other causes (NC-ARDS). The EIT exam was feasible in all included patients on veno-venous extracorporeal support. Overall, bedside respiratory mechanics, EIT regional ventilation distribution, and PEEP setting according to EIT did not differ significantly in these populations. Patients with severe NC-ARDS showed a higher risk of overdistension at higher PEEP level compared with patients with C-ARDS.

Early reports suggested that C-ARDS presents specific characteristics compared with the classical ARDS pattern [22,23] and has time-related characteristic patterns [24]. Specifically, in the early phases of the disease, it often presents with severe hypoxemia despite a near normal compliance of the respiratory system. Recent studies have reported two main ARDS phenotypes in these patients. The type L phenotype is characterized by low elastance (i.e., high compliance), low ventilation–perfusion ratio, and low recruitability [24]. Patients presenting with the type L phenotype may remain stable for a time and then either improve or worsen toward the type H phenotype, characterized by high elastance, high pulmonary shunt, and high recruitability [19], with a clinical presentation more similar to typical ARDS. The distinction between these two phenotypes may be challenging at the bedside, as the potential for lung recruitment presents large variability and the same disease often presents with significant non-uniformity. In this context, EIT represents a feasible and promising tool to study lung homogeneity and to estimate the degree of overdistension and de-recruitment.

Our study population included patients with severe C-ARDS on extracorporeal support characterized by low compliance and high pulmonary shunt and, hence, is more representative of the type H phenotype. This explains the similar respiratory characteristics between the two groups. Both populations were ventilated with similar “ultra-protective” ventilation strategies, and basal PEEP levels were not significantly different between the two groups. The basal PaCO_2_ level was high in the severe C-ARDS group, reflecting the underlying mechanism of altered perfusion and the structural changes in lung parenchyma associated with increased dead space [23]. 

Our results are consistent with recent studies that did not find remarkable differences in respiratory system compliance and parameters between the C-ARDS and NC-ARDS groups [25,26,27,28]. C-ARDS has pathophysiological features similar to non-COVID-19 ARDS, with reduced respiratory system compliance, high heterogeneity of respiratory mechanics, hypoxemia severity, and lung recruitability. There is no evidence supporting COVID-19-specific ventilatory settings, and the vast amount of available literature suggests that evidence-based, lung-protective ventilation (i.e., tidal volume ≤ 6 mL/kg, plateau pressure ≤ 30 cm H_2_O) should be enforced in all patients with COVID-19 ARDS who were mechanically ventilated. 

Overall, our study emphasizes that ventilation should be tailored to the patient respiratory mechanics rather than only on ARDS aetiology. 

Respiratory parameters such as arterial oxygen tension, compliance, and driving pressure did not differ after PEEP change in either group. However, global measures of oxygenation or respiratory system mechanics may produce discrepant information by “averaging” opposite pathological phenomena (i.e., tidal recruitment and overdistension) in different lung units.

The EIT-derived results provided a more detailed analysis and unveiled that, overall, de-recruitment and overdistension occur even for minimal PEEP changes compared with PEEP post, which corresponds to the best compromise between alveolar overdistension and de-recruitment. This finding is particularly relevant in severe ARDS, as PEEP increase might not lead to alveolar recruitment but might determine overdistension, hyperinflation, and lung damage [29,30]. The C-ARDS population was characterized by a slightly lower dorsal distribution of tidal volume, but this result might have been influenced by the higher amount of adipose tissue surrounding the chest wall in this specific group of patients. Our findings demonstrate that the two populations have similar respiratory characteristics, even if patients with severe NC-ARDS patients higher risk of overdistension and hyperinflation at higher PEEP levels.

Our results are consistent with recent literature. Franchineau et al. [11] previously demonstrated the feasibility of EIT for ventilatory monitoring of patients with ARDS on V-V ECMO support. In their study, high PEEP levels were associated with more overdistension (up to 50% of lung volume) and lower lung collapse, whereas decreasing PEEP led to a degree of de-recruitment as high as 72% of lung volume. Perier and colleagues [9] applied EIT to titrate PEEP in ARDS due to COVID-19 and other causes. The optimal PEEP using EIT in C-ARDS was 12 cm H_2_O [range 9–13.5], with de-recruitment at lower PEEP levels.

The main limitations of our study are mainly its retrospective nature and the small study population, with the consequent risk of low power. At the same time, the data appear very similar among the two populations so that a difference, even if present, would unlikely be clinically relevant. Due to the retrospective nature of the study, respiratory parameters after 24–48 h from the EIT exam were not collected and we could not assess if our findings were stable over time. Nevertheless, the results would have been influenced by other confounding factors such as pronation or would switch to assisted ventilation.

Another limitation may be related to the higher heterogeneity of the NC-ARDS group, which included pulmonary ARDS due to different aetiologies. Moreover, our population included only supine patients and did not investigate the potential clinical utility of EIT in understanding the physiologic dynamics of prone positioning [31], which would be of great interest in the ECMO population, as the benefit of this procedure during ECMO is still debated [32,33,34]. Furthermore, we did not consider changes in regional ventilation distribution during spontaneous breathing and the EIT role in guiding patients who had difficulty weaning [35].

Moreover, we recognise that a methodological limitation of our study relies on the fact that recruitment manoeuvres were performed at 35 cm H_2_O, which may not achieve a significant recruitment effect in patients with severe ARDS. However, this approach was preferred to limit the risk of barotrauma and hemodynamic impairment.

Lastly, the EIT-guided PEEP values did not overly diverge from the clinically based setting in our tertiary-level ICU specialized in ARDS treatment, but these findings may not apply to less experienced centres.

## 5. Conclusions

Our study demonstrated that EIT is feasible in patients with COVID-19-associated ARDS on veno-venous extracorporeal support and may help to tailor the PEEP setting. EIT regional ventilation analysis showed that C-ARDS seems to be associated with a lower risk of overdistension at higher PEEP levels compared with NC-ARDS. Overall, our study shows that severe COVID-19-related ARDS presents respiratory characteristics similar to severe “classical” NC-ARDS, supporting the fact that mechanical ventilation should be tailored to patient respiratory mechanics rather than to ARDS aetiology. 

## Figures and Tables

**Figure 1 jcm-11-01639-f001:**
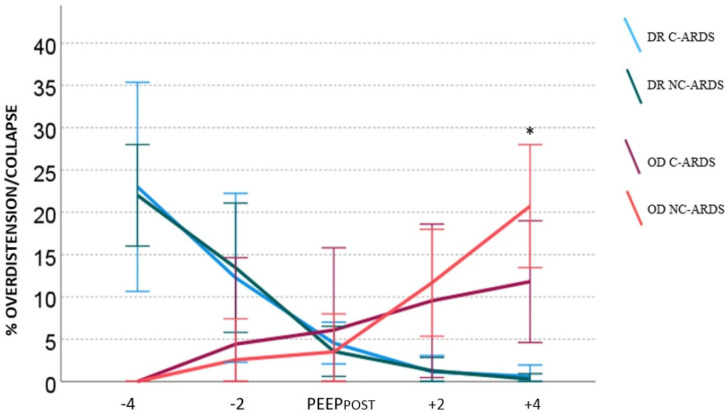
Overdistension (OD) and de-recruitment (DR) at different PEEP levels in the C-ARDS and NC-ARDS groups. Data are synthesized as means, and error bars represent standard deviation. * OD: *p* < 0.05 versus C-ARDS.

**Table 1 jcm-11-01639-t001:** Demographics, clinical characteristics, and outcome.

Population	C-ARDS	NC-ARDS
	(*n* = 12)	(*n* = 12)
Age, years	55 [43.5–60]	41 [31–59]
Sex (male)	7 (58%)	7 (58%)
BMI, kg/m^2^	33 [31–35]	28 [21–30] *
Total Invasive ventilation duration, days	25 [17–39]	42 [18–51]
Total ECMO duration, days	14 [9–25]	18 [13–32]
ICU length of stay, days	27 [17–35]	42 [23–53]
ICU mortality	1 (8%)	3 (25%) *
Hospital mortality	1 (8%)	3 (25%) *

Data are expressed as median [interquartile range] or absolute (relative) frequency. BMI, body mass index; ARDS, acute respiratory distress syndrome; ICU, intensive care unit; *n* = number of patients. * *p* ≤ 0.05.

**Table 2 jcm-11-01639-t002:** Clinical characteristics and ventilation parameters before EIT evaluation.

	C-ARDS (*n* = 13)	NC-ARDS (*n* = 18)
Pulmonary shunt fraction, %	48.5 [35–57]	53 [45–65]
Compliance of the respiratory system, mL/cm H_2_O	25 [22–33]	22 [15–29]
PEEP, cm H_2_O	14 [12–15.5]	15 [12–16.5]
Respiratory rate, breaths per minute	10 [10–10]	10 [10–10]
Tidal Volume, mL	250 [200–345]	205 [170–270]
Tidal Volume, mL/kg	4 [3.7–5]	3 [2.5–5]
Driving Pressure, cm H_2_O	10 [8–10]	9 [8–11]

Data are expressed as median [interquartile range] or absolute (relative) frequency; PEEP, positive end expiratory pressure.

**Table 3 jcm-11-01639-t003:** Arterial blood gases, respiratory parameters, and regional distribution of ventilation before and after the EIT exams.

	C-ARDS		NC-ARDS	
	(*n* = 13)		(*n* = 18)	
**Arterial Blood Gas**	**Before EIT**	**After EIT**	**Before EIT**	**After EIT**
pH	7.41 [7.36–7.44]	7.41 [7.39–7.43]	7.42 [7.39–7.44]	7.41 [7.37–7.43]
PaO_2_	86 [72–96]	86.6 [80–95]	76 [66–79]	79 [70–94]
PaCO_2_	53 [47–56]	52 [48–57]	46 [41–54.5]	45.5 [44–57]
**Respiratory parameters**	**Before EIT**	**After EIT**	**Before EIT**	**After EIT**
PEEP	14 [12–15.5]	14 [12.5–16]	15 [12–16.5]	15 [12–16]
Cpl, mL/cm H_2_O	25 [22–33]	27 [21.5–34.5]	22 [15–29]	25 [17–41]
Driving Pressure, cm H_2_O	10 [8–10]	10 [8–10.5]	9 [8–11]	9 [8–10]
Plateau Pressure, cm H_2_O	23 [22–25]	23 [22–25]	24 [22–27]	24 [22–25]
Tidal Volume, mL	250 [200–345]	250 [200–345]	205 [170–270]	205 [170–270]
**Regional ventilation**	**Before EIT**	**After EIT**	**Before EIT**	**After EIT**
TV % ROI 1 ventral	8 [5.5–14]	8 [5–13.5]	15 [11–24.5] *	15 [10–21.5] *
TV % ROI 2 middle-ventral	46 [39.5–57.5]	47 [38–57.5]	43 [39–46]	44 [41–46]
TV % ROI 3 middle-dorsal	37 [27.5–49.5]	42 [25–49]	33 [23–41]	33 [23–41]
TV % ROI 4 dorsal	4 [3–6]	4 [3–6]	6 [4–9] *	5.5 [4–8]

PaO_2_, arterial oxygen tension; PaO_2_/FiO_2_, PaO_2_ to fraction of inspired oxygen; PaCO_2_, arterial carbon dioxide tension; TV, tidal volume; ROI, region of interest. * *p* ≤ 0.05 vs. C-ARDS.

## Data Availability

The data presented in this study are available from the corresponding author upon request. The data are not publicly available due to privacy and ethical issues.

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
