# Peer review of "Bedside Selection of Positive End Expiratory Pressure by Electrical Impedance Tomography in Patients Undergoing Veno-Venous Extracorporeal Membrane Oxygenation Support: A Comparison between COVID-19 ARDS and ARDS from Other Etiologies"

_jcm, 2022, doi:10.3390/jcm11061639_

Round 1
Reviewer 1 Report
Dr. Di Pierro and colleagues present a retrospective clinical analysis evaluating the use of electrical impedance tomography to select the level of PEEP in a cohort of C-ARDS patients comparing it with a cohort of NC-ARDS patients.
The study addresses a relevant clinical topic as individualizing the level of PEEP is becoming an important and still not well-established aspect of lung protective ventilation in ARDS patients and even more so in those with most severe forms requiring support with ECMO. EIT is a lung functional imaging monitoring technique that provides bedside information to optimize lung protective ventilation settings and is particularly useful to individualize the level of PEEP.
Major comments
My major concern with this study is the lack of real novel information. The feasibility (the first statement of the author’s conclusions) and use of EIT in ARDS patients on ECMO has already been described so that the new addition is the specific use in COVID patients needing ECMO. After all the discussions, in my view somewhat artificial, regarding the ARDS nature or not of the COVID related respiratory failure and the yet to be proven phenotypes, the authors once more nicely show that there are many more similarities than differences between C-ARDS and NC-ARDS.
In addition, there are several methodological aspects that lessen the relevance of the information provided by this report.
First a decremental PEEP trial is based on the concept of lung recruitment aimed at minimizing the amount of lung collapse. This sort of lung volume normalization is important as it is essential to correctly interpret the mechanical behavior of the lung during the trial. With minimum or no collapse at the first PEEP steps, reductions in compliance can correctly been attributed to mainly a reduction in overdistension, and when reaching a maximum value of compliance further decreases to the onset of dependent lung collapse. In this study the authors performed an insufficient lung homogenization maneuver with 30 cmH2O which is known to have a minimum to no effect on lung recruitment especially in sick ARDS lungs (please see the classic studies by Rothen and G. Hedenstierna and most recent by Borges et al, and De Matos et al). Then, the observed changes in lung mechanics will always be a mixture between decreased overdistension and the effects of ongoing lung collapse which makes interpretation difficult, and the correct use of the method as originally described by Costa et al questionable. This may explain the lack of differences in PEEP and any other variables observed. I see this as a methodological information that should be discussed and specifically addressed.
Second, PEEP post is ill-defined. It is said to be set according to clinical judgment and EIT interpretation. This is confusing. What was this clinical judgement based on? If EIT is providing the best compromise between minimal overdistension and collapse a quite objective criterion this is the one that should be used. Otherwise what are the authors exactly comparing? PEEP post must be clearly defined and established.
Third, it is not stated whether lungs were recruited again after the trial and then left with the identified PEEPpost. This is important for adequate comparison of PEEPpre and post and the intrinsically related to the principle of a decremental PEEP titration.
If this was not done it would again have influenced the results.
Fourth it is important to know when mechanics and lung condition was assessed after PEEPpost selection trial? Immediately? Later? When?. Was this PEEP and the effects maintained?
I think these important shortcomings can explain the lack of differences in PEEP or other variables observed in this study.
Fith, the comparison group NC-ARDS could be considered a sort of control in this study that has some similarities with a retrospective case-control design. However, it is not clear how patients were chosen. It is stated that 12 out of 44 patients on ECMO of a previous study were selected for comparison. Selected how from the sample of 44? Based on which criteria? This needs to be clearly explained and reported.
Figure 1 it would be clearer to express the results in % overdistension and collapse instead of % compliance loss. The method if I am not mistaken is analyzing the mechanical behavior of the individual pixels/voxelxs of the EIT image so that it represents the % of the total (that is of the lung) that is presenting the behavior i.e overdistension or collapse. Please check.
Table 3: please add VT values and Pplat values. It will help to better understand why NC-ARDS patients have a lower driving pressure but a lower compliance than C-ARDS patients.
Author Response
Reviewer 1
Dr. Di Pierro and colleagues present a retrospective clinical analysis evaluating the use of electrical impedance tomography to select the level of PEEP in a cohort of C-ARDS patients comparing it with a cohort of NC-ARDS patients.
The study addresses a relevant clinical topic as individualizing the level of PEEP is becoming an important and still not well-established aspect of lung protective ventilation in ARDS patients and even more so in those with most severe forms requiring support with ECMO. EIT is a lung functional imaging monitoring technique that provides bedside information to optimize lung protective ventilation settings and is particularly useful to individualize the level of PEEP.
Major comments
My major concern with this study is the lack of real novel information. The feasibility (the first statement of the author’s conclusions) and use of EIT in ARDS patients on ECMO has already been described so that the new addition is the specific use in COVID patients needing ECMO. After all the discussions, in my view somewhat artificial, regarding the ARDS nature or not of the COVID related respiratory failure and the yet to be proven phenotypes, the authors once more nicely show that there are many more similarities than differences between C-ARDS and NC-ARDS.
- We completely agree with the Reviewer. We do believe that “there are many more similarities than differences between C-ARDS and NC-ARDS”. Our idea was to show that (despite a lot of recent literature) ARDS from COVID-19 is very similar to classical ARDS, and to underline that ventilation should be tailored on the patient respiratory mechanics and not on ARDS etiology. Discussion was edited to better describe this view.
In addition, there are several methodological aspects that lessen the relevance of the information provided by this report.
First a decremental PEEP trial is based on the concept of lung recruitment aimed at minimizing the amount of lung collapse. This sort of lung volume normalization is important as it is essential to correctly interpret the mechanical behavior of the lung during the trial. With minimum or no collapse at the first PEEP steps, reductions in compliance can correctly been attributed to mainly a reduction in overdistension, and when reaching a maximum value of compliance further decreases to the onset of dependent lung collapse. In this study the authors performed an insufficient lung homogenization maneuver with 30 cmH2O which is known to have a minimum to no effect on lung recruitment especially in sick ARDS lungs (please see the classic studies by Rothen and G. Hedenstierna and most recent by Borges et al, and De Matos et al). Then, the observed changes in lung mechanics will always be a mixture between decreased overdistension and the effects of ongoing lung collapse which makes interpretation difficult, and the correct use of the method as originally described by Costa et al questionable. This may explain the lack of differences in PEEP and any other variables observed. I see this as a methodological information that should be discussed and specifically addressed.
- We thank the Reviewer for his insightful comment. As described in the methods section, the RM was performed at 35 (not 30) cmH2O for 30 seconds. We do agree with the Reviewer that in severe ARDS patients this pressure may not allow to achieve complete lung recruitment. However, to limit patient risks, we decided not to use higher pressure regimens for RMs, as these might be associated with adverse events such as barotrauma and hemodynamic impairment (see for ex. Cavalcanti et al NEJM 2017). We added this relevant aspect in the revised discussion section, as a study limitation.
Second, PEEP post is ill-defined. It is said to be set according to clinical judgment and EIT interpretation. This is confusing. What was this clinical judgement based on? If EIT is providing the best compromise between minimal overdistension and collapse a quite objective criterion this is the one that should be used. Otherwise what are the authors exactly comparing? PEEP post must be clearly defined and established.
- We thank the Reviewer for his comment, which allows us to make useful elucidations. As the Reviewer wrote, the EIT exam provides a “best PEEP level”, which is the best compromise between overdistension and derecruitment based on EIT findings. However, as described in the revised methods section, when the EIT findings did not allow to univocally decide between two PEEP levels, PEEPPOST setting was left to the decision of the clinician, who integrated EIT exam results and other patient information such as hemodynamic and oxygenation status. We have, accordingly, revised the methods section to clarify this point.
Third, it is not stated whether lungs were recruited again after the trial and then left with the identified PEEPpost. This is important for adequate comparison of PEEPpre and post and the intrinsically related to the principle of a decremental PEEP titration. If this was not done it would again have influenced the results.
- We thank the Reviewer for this relevant comment. A recruitment manoeuvre (35cmh2O, 30 sec) followed the lowest step of each decremental peep trial. This was added to the revised methods section.
Fourth it is important to know when mechanics and lung condition was assessed after PEEPpost selection trial? Immediately? Later? When?. Was this PEEP and the effects maintained? I think these important shortcomings can explain the lack of differences in PEEP or other variables observed in this study.
- We thank the Reviewer for his comment that helps us clarify this point. We assessed respiratory mechanics 30 minutes after the end of the study and selection of PEEPPOST. It would have been interesting to explore if the effects on lung mechanics were stable over time however, due to the retrospective nature of our study, we decided not collect data on respiratory mechanics and other parameters 24- 48 hours after the exam, as these findings could have been influenced by other confounding factors (e.g. pronation, switch to assisted ventilation, etc). The study limitations section now includes this consideration.
Fith, the comparison group NC-ARDS could be considered a sort of control in this study that has some similarities with a retrospective case-control design. However, it is not clear how patients were chosen. It is stated that 12 out of 44 patients on ECMO of a previous study were selected for comparison. Selected how from the sample of 44? Based on which criteria? This needs to be clearly explained and reported.
- We thank the reviewer for highlighting this point. The total subset of 41 patients from the previous study included 41 patients who received an EIT exam with various approaches (incremental or decremental peep trial, sigh setting) during the determined time interval. Out of them, 18 received a decremental PEEP trial and only 12 were on ECMO support due to ARDS of pulmonary aetiology. To homogenize the study population, we chose to limit our analysis to ECMO patients, who shared life threatening impairment of lung function and gas exchange. We have amended the revised manuscript to be clearer.
Figure 1 it would be clearer to express the results in % overdistension and collapse instead of % compliance loss. The method if I am not mistaken is analyzing the mechanical behavior of the individual pixels/voxelxs of the EIT image so that it represents the % of the total (that is of the lung) that is presenting the behavior i.e overdistension or collapse. Please check.
- We thank the Reviewer for the comment. Figure 1 was edited according to the Reviewer correction, replacing the Y axis label with “% of overdistention/collapse”.
Table 3: please add VT values and Pplat values. It will help to better understand why NC-ARDS patients have a lower driving pressure but a lower compliance than C-ARDS patients.
- As per Reviewer comment, TV and Pplat values were added to Table3. NC-ARDS patients had a lower driving pressure despite a lower compliance because the median TV was lower (200 vs 250).
Reviewer 2 Report
General comments:
This paper by Di Pierro and colleagues retrospectively describe the results of EIT assessments performed in Covid-ARDS and No-Covid-ARDS patients undergoing V-V ECMO support. A clinical EIT-guided decremental PEEP trial was conducted to all ventilated patients included mechanically. 12 C-ARDS and 12 NC-ARDS patients were included in the study for a total of 13 and 18 EIT evaluations, respectively. No significant differences in arterial blood gas, respiratory parameters and regional ventilation before and after the EIT exam were recorded. The subset of NC-ARDS patients whose EIT exam led to PEEP modification was characterized by a lower baseline compliance compared to C-ARDS group: 18 [16-28] vs 27 [24-30] (p=0.04). Overdistension significantly increased at higher steps only for the NC-ARDS group. A higher percentage of over-distension was described in NC-ARDS patients when compared to C-ARDS patients. The authors concluded that EIT is feasible in patients with COVID-19 associated ARDS on veno-venous extracorporeal support and may help to tailor the PEEP setting. Overall, severe COVID-19 related ARDS presents respiratory characteristics similar to severe “classical” NC-ARDS. However, C-ARDS is associated to a lower risk of overdistension at higher PEEP level compared to NC-ARDS.
Specific comments:
- Page 3 line 98: How did the authors define the clinical PEEP. Please clarify…
- The authors included in their study 12 C-ARDS compared to 12 NC-ARDS. Why did the authors presented the results of 13 measurements of C-ARDS and 18 NC-ARDS in tables 2 and 3. Isn´t it better to show one result of each patient in terms of comparison. Please clarify and discuss.
- In the discussion section Pg 7 line 231 the authors stated `as PEEP increase might not lead to alveolar recruitment but determine overdistension, lung shear stress and the risk of hyperinflation and lung damage [29-30]. Lung shear stress is Much more a phenomenon related to colapse /normal lungs and high tidal volumes ( distending pressures ?) Isn`t it...please, discuss and clarify.
- Pg 7 line 247 The main limitations of our study mainly rely on its retrospective nature and the 247 small study population, with the consequent risk of underpower. Underpower for what? Please clarify .
Author Response
Reviewer 2
General comments:
This paper by Di Pierro and colleagues retrospectively describe the results of EIT assessments performed in Covid-ARDS and No-Covid-ARDS patients undergoing V-V ECMO support. A clinical EIT-guided decremental PEEP trial was conducted to all ventilated patients included mechanically. 12 C-ARDS and 12 NC-ARDS patients were included in the study for a total of 13 and 18 EIT evaluations, respectively. No significant differences in arterial blood gas, respiratory parameters and regional ventilation before and after the EIT exam were recorded. The subset of NC-ARDS patients whose EIT exam led to PEEP modification was characterized by a lower baseline compliance compared to C-ARDS group: 18 [16-28] vs 27 [24-30] (p=0.04). Overdistension significantly increased at higher steps only for the NC-ARDS group. A higher percentage of over-distension was described in NC-ARDS patients when compared to C-ARDS patients. The authors concluded that EIT is feasible in patients with COVID-19 associated ARDS on veno-venous extracorporeal support and may help to tailor the PEEP setting. Overall, severe COVID-19 related ARDS presents respiratory characteristics similar to severe “classical” NC-ARDS. However, C-ARDS is associated to a lower risk of overdistension at higher PEEP level compared to NC-ARDS.
Specific comments:
Page 3 line 98: How did the authors define the clinical PEEP. Please clarify.
- We thank the reviewer for the comment, which allow us to clarify this point. “Clinical peep” was the basal peep value already set by clinicians before the EIT exam. In our routine clinical practice, optimal PEEP value is usually set based at a level which provides the best respiratory system compliance and oxygenation. For clarity, the sentence “clinical PEEP” was rephrased as follows “ pre-existing PEEP level”.
The authors included in their study 12 C-ARDS compared to 12 NC-ARDS. Why did the authors presented the results of 13 measurements of C-ARDS and 18 NC-ARDS in tables 2 and 3. Isn´t it better to show one result of each patient in terms of comparison. Please clarify and discuss.
- We thank the reviewer for the comment that allow us to clarify this point. We decided to take into consideration the total amount of EIT exams since our primary goal was to describe the role of EIT in helping and guiding physicians in the ventilator strategy of ARDS patients. The EIT exam was always performed after physicians demand and in some cases, it was requested more than once and at different time interval. We opted for including all the examinations, as we believed it was important to describe the findings related to the different clinical phases of the disease.
In the discussion section Pg 7 line 231 the authors stated `as PEEP increase might not lead to alveolar recruitment but determine overdistension, lung shear stress and the risk of hyperinflation and lung damage [29-30]. Lung shear stress is Much more a phenomenon related to colapse /normal lungs and high tidal volumes ( distending pressures ?) Isn`t it...please, discuss and clarify.
- We understand the reviewer’s viewpoint and we apologise for any confusion. To avoid any misconception, the expression shear stress was removed and the sentence was rephrased as follows: “ but determine overdistension, hyperinflation and lung damage”.
Pg 7 line 247 The main limitations of our study mainly rely on its retrospective nature and the 247 small study population, with the consequent risk of underpower. Underpower for what? Please clarify.
- We thank the reviewer for raising this important point. A formal sample size estimation was not possible due to the nature of the study (retrospective) and the lack of adequate data in literature to estimate a difference between groups. We wanted to disclose that with a relatively small population the chance of detecting a statistically significant difference might be reduced. We considered our sample size as a limitation of our study as it might not be large enough to answer the research question of interest. We cannot exclude that a larger study population might have uncovered small differences between the two populations.
Round 2
Reviewer 1 Report
I thank the authors for the revised version of the manuscript. They have addressed most of my concerns. I have however some remaining comments.
The lack of novelty of has not been convincingly addressed. I still think this study is not providing any relevant new information.
Table 3: the legend indicates that significance was just assessed between C-ARDS and NC-ARDS. It is strange that authors did not compare the before and after EIT moments for each group, at least this is not clear in the text. Why? Were there no differences at all in the studied variables after EIT PEEP adjustment? I think it is important to add this analysis and information even if there were no differences, which then would logically raise some questions regarding the usefulness of EIT to optimize /individualize mechanical ventilation at least in the way it is proposed by the authors.
Table 3 it is surprising that with a lower VT and a higher PEEP the PaCO2 levels are numerically lower (when I would be expect them to be higher) in the NC-ARDS group. Any explanation?
Table 3: I understand that the = in the VT column means equal value. I would rather see the numerical values here as this may be confusing (for instance it is not mentioned in the legend).
During an EIT guided decremental PEEP trial the so called “crossing point” i.e the PEEP level indicating the lowest overdistension and lowest lung collapse should always be posible to detect by the very nature of the measurement principle. What is meant by “when EIT findings did not allow to unequivocally (please also correct the typo as authors wrote univocally) decide between two PEEP levels..”? this is unclear. Was it because of an irregular recording or a noisy signal? . Were there any quality criteria for the EIT recording used? Could you show an example? In how many patients did this happen? PEEP is the principal study variable and introducing the bias of clinical judgment is a very important limitation for the asessment of the goodness/usefulness of the method. In fact there were no changes in the overall levels of PEEP. This needs more clarification and described as a limitation.
Page 8 discussion: line 283: relies.
Next lines authors state: “ may not allow to achieve full lung recruitment”. It would be more correct to write: “may not achieve a significant recruitment effect”. The recruitment effect of different opening pressures has been studied in detail by Borges et al and De Matos et al. Full lung recruitment as assessed by CT was achieved in roughly 45% of the patients when applying 40 cmH2O of pressure (see Borges et al Am J Respir Crit care Med 2006;174:268-278) so that the recruitment effect of the protocol used was probably quite low. In my first review I already discussed how this affects the interpretation of the decremental PEEP trial.
Page 8 line 241- “emphasizes”
Page 3 line 116: unequivocally
Author Response
I thank the authors for the revised version of the manuscript. They have addressed most of my concerns. I have however some remaining comments.
The lack of novelty of has not been convincingly addressed. I still think this study is not providing any relevant new information.
- We sincerely thank the author for the attentive reply and we understand his point. Plentiful recent literature is available on EIT, also on ECMO patients. We think the original aspect of our study is to compare EIT findings between Covid and Non-Covid ARDS, in the particular subset of patients with severe ARDS on ECMO. To the best of our knowledge, this comparison has not been performed in this population to date. In our eyes, our study is important to support the idea that there is no COVID-19-specific ventilatory setting, and underlines the fact that ventilation should not be based on ARDS aetiology but on patient respiratory mechanics and regional lung characteristics.
Table 3: the legend indicates that significance was just assessed between C-ARDS and NC-ARDS. It is strange that authors did not compare the before and after EIT moments for each group, at least this is not clear in the text. Why? Were there no differences at all in the studied variables after EIT PEEP adjustment? I think it is important to add this analysis and information even if there were no differences, which then would logically raise some questions regarding the usefulness of EIT to optimize /individualize mechanical ventilation at least in the way it is proposed by the authors.
- We thank the Reviewer for the useful question. We compared the variables before and after the EIT exam and we did not find significant differences. This finding is presented in the results section (line 169). However, we did not emphasize the comparison before vs after EIT evaluation because this was not the goal of the study. Furthermore, PEEP change after the EIT exam was not significant, thus major changes in respiratory parameters couldn’t be expected either.
Table 3 it is surprising that with a lower VT and a higher PEEP the PaCO2 levels are numerically lower (when I would be expect them to be higher) in the NC-ARDS group. Any explanation?
- We thank the Reviewer for the question which allows us to make a useful clarification. We think the elevated PaCO2 level in severe C-ARDS group may correlate to the underlying mechanism of altered perfusion and the structural changes in lung parenchyma associated with increased alveolar dead space. These aspects appear typical of the Covid disease pathogenesis itself. However, we recognize the limit of PaCO2 interpretation in ECMO patients, as it is greatly influenced by the sweep gas flow level.
Table 3: I understand that the = in the VT column means equal value. I would rather see the numerical values here as this may be confusing (for instance it is not mentioned in the legend).
- We thank the Reviewer for the correction. We added the values in the table.
During an EIT guided decremental PEEP trial the so called “crossing point” i.e the PEEP level indicating the lowest overdistension and lowest lung collapse should always be posible to detect by the very nature of the measurement principle. What is meant by “when EIT findings did not allow to unequivocally (please also correct the typo as authors wrote univocally) decide between two PEEP levels”? this is unclear. Was it because of an irregular recording or a noisy signal? . Were there any quality criteria for the EIT recording used? Could you show an example? In how many patients did this happen? PEEP is the principal study variable and introducing the bias of clinical judgment is a very important limitation for the asessment of the goodness/usefulness of the method. In fact there were no changes in the overall levels of PEEP. This needs more clarification and described as a limitation.
- We thank the reviewer for the comment and we apologize for the misunderstanding. We used the “crossing point” at the EIT final exam as the reference point to define the PEEPeit. In two cases (1 for the CARDS group and 1 for the NCARDS) clinicians did not recognize an explicit benefit between two PEEP values (either very minimal difference between two values or crossing point only very slightly moved toward one side of the 2cmH20 step). Unfortunately, due to the retrospective nature of the study, we could not avoid this limitation. We recognize that this concept may be confusing and we modified the explanation on the text.
We did not met any quality problems or irregularities in the EIT recording. We set the recording at 50Hz and the electrodes signal was always checked and classified at least as “good”.
Page 8 discussion: line 283: relies.
- We thank the Reviewer for the correction.
Next lines authors state: “ may not allow to achieve full lung recruitment”. It would be more correct to write: “may not achieve a significant recruitment effect”. The recruitment effect of different opening pressures has been studied in detail by Borges et al and De Matos et al. Full lung recruitment as assessed by CT was achieved in roughly 45% of the patients when applying 40 cmH2O of pressure (see Borges et al Am J Respir Crit care Med 2006;174:268-278) so that the recruitment effect of the protocol used was probably quite low. In my first review I already discussed how this affects the interpretation of the decremental PEEP trial.
- We thank the Reviewer for the suggestion, discussion was edited accordingly. We agree with the Reviewer that higher pressure recruitment maneuver are more effective in achieving lung recruitment. However, we preferred to use lower pressure regimens in order to prevent the haemodynamic and barotraumatic side effects, as explained also in the above-mentioned study. Moreover, as already described (see Gattinoni et al Eur Resp Rev 2021), with the increasing severity of the disease the response to recruitment maneuver might progressively vanish, whereas the risk of side effect may increase.
Page 8 line 241- “emphasizes”
- Text was edited according to the Reviewer suggestion.
Page 3 line 116: unequivocally
- We thank the Reviewer for the correction. We edited the text.